# Chemical Constituents and Antimicrobial Activity of a *Ganoderma lucidum* (Curtis.) P. Karst. Aqueous Ammonia Extract

**DOI:** 10.3390/plants12122271

**Published:** 2023-06-11

**Authors:** Eva Sánchez-Hernández, Ana Teixeira, Catarina Pereira, Adriana Cruz, Jesús Martín-Gil, Rui Oliveira, Pablo Martín-Ramos

**Affiliations:** 1Department of Agricultural and Forestry Engineering, ETSIIAA, University of Valladolid, Avenida de Madrid 44, 34004 Palencia, Spain; mgil@iaf.uva.es (J.M.-G.); pmr@uva.es (P.M.-R.); 2Department of Biology, School of Sciences, University of Minho, Campus de Gualtar, 4710-057 Braga, Portugal; anaspereirateixeira@gmail.com (A.T.); catarinavalepereira@gmail.com (C.P.); cruzadriana73@gmail.com (A.C.); ruipso@bio.uminho.pt (R.O.); 3Centre of Molecular and Environmental Biology (CBMA), University of Minho, Campus de Gualtar, 4710-057 Braga, Portugal

**Keywords:** antifungal activity, anti-oomycete activity, chitosan oligomers (COS), dehesa ecosystem, gas chromatography–mass spectrometry (GC-MS), mushroom extracts, natural products, phytopathogens, *Quercus ilex*, reishi

## Abstract

Mushroom extracts have shown potential as a source of new antimicrobial agents. This study investigates the chemical profile of an aqueous ammonia extract obtained from the carpophores of *Ganoderma lucidum*, which grows on *Quercus ilex* trees, and explores its valorization as a biorational. The major chemical constituents of the extract, identified through gas chromatography–mass spectrometry, include acetamide, oleic acid, 1,2,3,4-butanetetrol, monomethyl azelate, undecane, and palmitic acid. The anti-oomycete and antifungal activity of *G. lucidum* extract was evaluated against *Phytophthora cinnamomi*, the primary threat to *Quercus* spp. in the *dehesa* biome, as well as three Botryosphaeriaceae fungi. In vitro tests revealed minimum inhibitory concentration (MIC) values of 187.5 μg·mL^−1^ against *P. cinnamomi* and 187.5–1000 μg·mL^−1^ against the fungi. Furthermore, conjugation of the *G. lucidum* extract with chitosan oligomers (COS) synergistically enhanced its antimicrobial activity, resulting in MIC values of 78.12 and 375–500 μg·mL^−1^ against *P. cinnamomi* and the fungi, respectively. These MIC values are among the highest reported to date for natural products against these phytopathogens. Subsequent ex situ testing of the COS-*G. lucidum* conjugate complex on artificially inoculated *Q. ilex* excised stems resulted in high protection against *P. cinnamomi* at a dose of 782 µg·mL^−1^. These findings support the potential utilization of this resource from the *dehesa* ecosystem to protect the holm oak, aligning with sustainable and circular economy approaches.

## 1. Introduction

Medicinal mushrooms’ fruiting bodies, mycelium, and spores are valuable sources of bioactive products [1]. *Ganoderma lucidum* (Curtis.) P. Karst. is a dark, large fungus with a glossy exterior and a woody texture. It has been used for promoting health and longevity in Japan and China, where it is known as ‘reishi’ or ‘mannentake’, and ‘lingzhi’, respectively. The *G. lucidum* fruiting body has a tawny-to-russet-colored stipe (Figure 1). The context tissue, cinnamon-buff to pink-buff in color, shows concentric growth zones.

Several researchers have carried out the extraction of metabolites from *G. lucidum* using various solvents, namely, methanol, chloroform, acetone, or water [2,3]. *Ganoderma lucidum* extracts contain secondary metabolites such as phenols, steroids, terpenoids, nucleotides, glycoproteins, and polysaccharides [4]. Polysaccharides (ganoderans) and triterpenes (ganoderic acids, ganodermanondiol, ganodermanontriol, ganolucidic acid B, and lucidumol B) are the major bioactive chemical constituents [5,6]. 

The biological activity of *G. lucidum* has been investigated by Mizuno et al. [7] and Liu et al. [8]. Its polysaccharide composition significantly contributes to *G. lucidum*’s immunomodulatory, antioxidant, antitumor, and antibacterial properties [5,9]. On the other hand, its triterpene content is responsible for its antitumor, anti-inflammatory, antioxidant, anti-hepatitis, antimalarial, hypoglycemic, antimicrobial, and anti-inflammatory activity [10,11]. Furthermore, its polyphenol content plays a role in its antioxidant, antimicrobial, and anti-inflammatory properties, as well as its anti-tyrosinase activity [12,13].

The antimicrobial activity of *G. lucidum* extracts has been evaluated against bacteria such as *Bacillus subtilis*, *Staphylococcus aureus*, *Klebsiella aerogenes*, *Corynebacterium diphtheriae*, *Escherichia coli*, *Salmonella* spp., and *Pseudomonas aeruginosa* [2], as well as against fungi such as *Aspergillus niger*, *Aspergillus fumigatus*, *Aspergillus flavus*, *Mucor indicus*, *Curvularia lunata*, *Fusarium oxysporum*, *Alternaria alternata*, *Drashelaria* spp., and *Penicillium* spp. [3]. Yang et al. [14] demonstrated that *G. lucidum* polysaccharides combined with small amounts of chemical fungicides were successful in controlling plant diseases such as wheat brood, root rot, and corn stalk rot.

Concerning phytopathogens, *Phytophthora* spp. are a threat to global food security and the health, function, and biodiversity of native ecosystems [15]. The *dehesa* (semi-natural open woodlands) is a characteristic ecosystem of the Iberian Peninsula that is affected by one of these pathogens. The loss of trees due to the disease caused by the oomycete *Phytophthora cinnamomi* is one of the most significant problems that this biome faces, which is exacerbated by climate change [16]. *Phytophthora cinnamomi* is a globally distributed pathogen that can infect thousands of species and is considered to be the main biotic driver of *Quercus* spp. woodlands’ decline in Spain [17]. It is also one of the most threatening invasive pathogens in the world [18]. In addition to *P. cinnamomi*, the trees in the *dehesa* are also threatened by ascomycete fungi of the genus *Botryosphaeria*, including *Botryosphaeria dothidea*, *Diplodia corticola*, and *Dothiorella iberica*. These fungi cause cankers and dieback of twigs and have been associated with the decay of holm oaks and cork oaks, although *B. dothidea* has also been found in other species of the genus *Quercus* such as *Quercus robur* L. and *Quercus rubra* Michx. L. [19].

Taking into consideration that the use of fungicides is discouraged under the current new European Union forest strategy for 2030 (Sustainable Forest Management in Europe, 2022/2016(INI)) and that Article 14 of Directive 2009/128/EC promotes the use of formulations based on natural ingredients as new protection techniques, the study presented herein aims to study the chemical constituents present in *G. lucidum* aqueous ammonia extract by gas chromatography–mass spectrometry (GC–MS) and to explore opportunities for the valorization of this extract for the control of aforementioned phytopathogens. This second goal was addressed by first studying the in vitro antifungal and anti-oomycete activity of the aqueous ammonia extract, alone and upon conjugation with chitosan oligomers (COS), and by subsequent ex situ testing of the most effective treatment on *Quercus ilex* L. excised stems to confirm its anti-oomycete activity against *P. cinnamomi*.

## 2. Results

### 2.1. Infrared Vibrational Characterization

The primary absorption bands in the infrared spectra of *G. lucidum* carpophores powder are summarized in Table 1, alongside those of the commercial *G. lucidum* powder of Chinese origin. The identified functional groups are consistent with the presence of the chemical constituents identified in the aqueous ammonia extract by GC−MS (such as polyphenols, esters of organic acids, and alkaloids), together with non-extracted constituents, such as glucans (a characteristic *β*-glucan band appears at 1036 cm^−1^ and another band which represents (1→4) linked glucans is located at 1153 cm^−1^).

### 2.2. Extract Phytoconstituents Elucidation by GC−MS

The main components of the *G. lucidum* carpophore aqueous ammonia extract (Appendix A and Figure 2, and Table 2) were: acetamide or ethanamide (28.3%); 9-octadecenoic acid and its methyl ester (8%); l-threitol (or 1,2,3,4-butanetetrol) (4.8%); nonanedioic acid, monomethyl ester (4.8%); undecane (4.5%); n-hexadecanoic acid (palmitic acid) and its methyl ester (4.6%); glycerin (3.9%); 2,6-dimethoxy-phenol (2.5%); 5-hydroxy-2(1H)-pyridinone (2.5%); mequinol or 4-hydroxyanisole (2.2%); N-methoxy-2-carbamino aziridine (2.2%); dodecanoic acid and its methyl ester (2.2%); 3-(acetyloxy)-N,N-dimethyl-2-propenethioamide (2.1%); and N,N-dimethylaceto acetamide (1.7%).

### 2.3. Antifungal and Anti-Oomycete Activity

The results of the antifungal/anti-oomycete susceptibility test are presented in Figure 3. An increase in concentration led to a decrease in the radial growth of the mycelium for all three tested products (COS, *G. lucidum* carpophore aqueous ammonia extract, and their conjugate complex), resulting in statistically significant differences. The aqueous ammonia extract of *G. lucidum* carpophores exhibited higher antifungal/anti-oomycete activity than COS, with minimum inhibitory concentrations (MICs) ranging from 187.5 to 1000 μg·mL^−1^ and from 750 to 1500 μg·mL^−1^, respectively. *Phytophthora cinnamomi* was the most sensitive phytopathogen in both cases, with MIC values of 187.5 and 750 μg·mL^−1^ for *G. lucidum* extract and COS, respectively. The formation of conjugate complexes improved the activity, with the COS–*G. lucidum* conjugate producing complete inhibition of *Botryosphaeriaceae* family pathogens at concentrations in the range of 375 to 500 μg·mL^−1^, while the inhibition value was as low as 78.12 μg·mL^−1^ for *P. cinnamomi*. The 50 and 90% effective concentrations (EC_50_ and EC_90_, respectively), presented in Table 3, allow for a clearer observation of this enhancement of the antifungal/anti-oomycete activity, which was quantified according to Wadley’s method. The synergy factor values were in the range of 1.98–3.63. As these values were higher than 1, a synergistic behavior can be inferred in all cases.

For the purpose of comparison, Fosetyl-Al, a conventional synthetic fungicide widely employed against *Phytophthora* spp. and fungi associated with grapevine trunk diseases (GTDs), was utilized as a positive control. As indicated in Table 4, when administered at the recommended dose of 2000 μg·mL^−1^ (equivalent to 2.5 g·L^−1^ for Fosbel^®^, fosetyl-Al 80%), complete inhibition of the four phytopathogens was observed. However, when applied at one-tenth of the recommended dose, a moderate inhibition was observed against *B. dothidea* and *D. corticola*, while a weak inhibition was observed in the case of *P. cinnamomi*, and no inhibition was detected against *D. iberica*. 

### 2.4. Protection of Excised Stems against P. cinnamomi

The COS−*G. lucidum* conjugate complex was the most active product in the in vitro tests and was subsequently tested as a protective treatment against *P. cinnamomi* on holm-oak-excised stems. Three different concentrations were used, corresponding to the MIC, MIC × 5, and MIC × 10 (i.e., 78, 391, and 782 µg·mL^−1^, respectively). Results are presented in Appendix A, and a comparison of canker lengths is shown in Table 5. No protective effect was observed at the lowest dose tested (i.e., at the MIC value obtained in the in vitro tests), with canker lengths similar to those of the positive control (non-treated stems infected with the oomycete). At a dose equal to five times the MIC, significantly lower canker lengths were observed. However, it was necessary to increase the concentration up to 10 times the MIC to achieve effective protection, with no significant differences compared to the negative control. Nevertheless, at this dose, small cankers were still visible in four of the excised stems (out of fifteen replicates), indicating that higher doses may be required in field conditions. 

## 3. Discussion

### 3.1. On the Chemical Profile

Among the list of compounds presented in Table 2, acetamide or ethanamide has been previously identified in red beetroots (*Beta vulgaris* var. *rubra*) and *Clerodendrum infortunatum* L. leaves [20]. It has also been found in extracts from *Larrea divaricata* Cav., *Picea pungens* Engelm., and *Sequoiadendron giganteum* (Lindl.) Buchholz. The presence of acetamide in the extract may be attributed to the partial hydrolysis of *N,N*-dimethylacetoacetamide, which was also identified in the extract. Alternatively, it could originate from N-(3-methylbutyl)acetamide or N(2-phenylethyl)acetamide, which are common components of fresh wild mushrooms [21]. However, it is worth noting that the presence of acetamide in the extract may be an artifact resulting from the extraction procedure, as it can also be formed through the decomposition of ammonium acetate. Ammonium acetate is generated by neutralizing excess ammonia in the extract with acetic acid. It is important to mention that acetamide-containing compounds are widely used as herbicides in agriculture [22], and several acetamide derivatives have been reported to act as antimicrobial agents [23].

Oleic acid, or 9-octadecenoic acid, has been identified in damask rose oil [24], *Chenopodium album* L. root methanolic extract [25], *Allium sativum* Regel L. [26], *Sesuvium portulacastrum* L. [27], *Armeria maritima* (Mill.) Willd. [28], *Taxus baccata* L. [29], and in small amounts in pomegranates, peas, cabbages [30], *Foeniculum vulgare* Mill. [31], and *Landolphia owariensis* Beauv. [32]. Its antifungal activity has been demonstrated against soil pathogens affecting the family Cucurbitaceae, namely, *Fusarium equiseti*, *Fusarium oxysporum* f. sp. *niveum*, *Neocosmospora falciformis*, *Neocosmospora keratoplastica*, *Macrophomina phaseolina*, and *Sclerotinia sclerotiorum* [28], corroborating the activity previously reported by Walters et al. [33] against *Crinipellis perniciosa*, a pathogen of the genera *Theobroma* and *Herrania*, responsible for witches’ broom, as well as against the oomycete *Pythium ultimum*, which affects flower bulbs, summer flowers, and perennials.

L-threitol, also known as 1,2,3,4-butanetetrol, is a non-cariogenic component found in Shiitake mushrooms [34] and is also the primary component of *Thaumatococcus daniellii* (Benn.) Benth. ex B.D.Jacks. leaves [35]. At present, there is no available information on the antimicrobial, antibacterial, or antifungal activity of this compound.

Nonanedioic acid (or 8-carbomethoxyoctanoic acid) monomethyl ester, also known as monomethyl azelate, is a dicarboxylic acid naturally produced by *Malassezia furfur* (C.P. Robin) Baill. and is also present in whole-grain cereals, rye, and barley. It is known to be effective in treating acne and various cutaneous disorders [36]. 

Undecane was previously identified as a major constituent of the extract of the stem bark of *Symplocos crataegoides* Buch.-Ham. ex D. Don (7.5%) [37], *Opuntia ficus indica* (L.) Mill (20%) [38], *Seseli pallasii* Besser (13.3%), *T. baccata* (12.2%) [29], and in smaller percentages in the essential oils of *Hypericum hirsutum* L. [39] and *Lantana camara* L. [40]. There is no clear information available on the mechanism of action of undecane as an antimicrobial agent.

n-Hexadecanoic acid (palmitic acid) and its methyl ester were identified in several plants such as *Equisetum arvense* L. (18.3%) [41], *A. maritima* (18%) [28], *Limonium binervosum* (G.E.Sm.) C.E. Salmon (15%) [42], *Hibiscus syriacus* L. (9.6%) [43], and *Tamarix gallica* L. (3.7%) [44]. Palmitic acid has been found to have nematicide and pesticide properties [45]. Moreover, it has demonstrated antifungal activity against various fungi, including *Alternaria solani, F. oxysporum*, *Colletotrichum lagenaria*, *A. niger*, *Aspergillus terreus*, *Aspergillus nidulans*, *N. falciformis*, *N. keratoplastica*, *M. phaseolina*, and *S. sclerotiorum* [28,46,47]. 

2,6-Dimethoxyphenol (syringol) has been identified in various extracts, including *Macrotermes gilvus* fungus combs (6.5%) [48], *Uncaria tomentosa* (Willd. ex Schult.) DC. [49], and *T. gallica* [44]. The antimicrobial effects of syringol isolated from *Camelia japonica* wood vinegar have been demonstrated against *Globisporangium splendens*, *Ralstonia solanacearum*, *F. oxysporum*, and *Phytophthora capsici* [50].

5-hydroxy-2(1H)-pyridinone is analogous to 6-hydroxy-2(1H)-pyridinone, the primary natural compound found in the wild berry *Rubus fraxinifolius* Poir. [51]. Although no information is currently available on the antimicrobial activity of 5-hydroxy-2(1H)-pyridinone, the 2(1H)-pyridone ring system is abundantly found in a wide variety of naturally occurring alkaloids and novel synthetic biologically active molecules. Heterocycles containing a 2(1H)-pyridone framework constitute a highly studied class of compounds due to their diverse biological activities, including anti-HIV, antibacterial, antifungal, and free radical scavengers [52].

### 3.2. On the Antimicrobial Activity Comparison of G. lucidum Extracts

The antibacterial and antifungal activity results reported for *G. lucidum* aqueous ammonia extract in this study are consistent with the previously reported antimicrobial activity of *G. lucidum* extracts in other solvents (Appendix A) [3,12,53,54,55,56,57,58,59]. However, previous reports have primarily focused on human pathogens, with limited data on phytopathogens, thus making a direct comparison among extraction media unfeasible.

### 3.3. Comparison of Efficacy vs. Other Natural Compounds

The use of different isolates with distinct susceptibility profiles generally makes it difficult to accurately compare the activity of *G. lucidum* aqueous ammonia extract with that of other plant extracts reported in the literature (see Table 6). Nevertheless, it can be observed that *G. lucidum*-based treatments exhibit some of the highest activities against the four phytopathogens. Regarding *B. dothidea*, the efficacy of the pure extract is comparable to that of a compound herbal extract compound consisting of seven Chinese medicinal plants [60]. Meanwhile, the activity of the conjugate complex is intermediate between those of COS-*U. dioica* and COS-*E. arvense* conjugates [41], tested against the same isolate. Concerning *D. corticola*, the extract displays the highest activity. As for *D. iberica*, the data are only available for COS-*U. dioica* and COS-*E. arvense* conjugates [41] (tested against the same isolate), which exhibited lower activity, with MIC values at least twice that of the COS-*G. lucidum* conjugate complex. In terms of the activity against *P. cinnamomi* (MIC = 187.5 for *G. lucidum* extract), it is only lower than those reported for an aqueous ammonia extract of holm oak bark (MIC = 78.12 µg·mL^−1^) [61] and *O. ficus-indica* aqueous extract (EC_90_ = 121.7 µg·mL^−1^) [62], and comparable to those of *Flourensia cernua* DC. extract (EC_90_ = 193.4 µg·mL^−1^) [62] and *Thymus vulgaris* L. essential oil (MIC = 200 µg·mL^−1^) [63].

### 3.4. Comparison of Efficacy vs. Fosetyl-Al

Upon comparing the values of mycelial growth inhibition for Fosetyl-Al (as shown in Table 4) with the effective concentrations reported for *G. lucidum* extract and its conjugate complexes (as presented in Table 3), it can be observed that the in vitro activity of the natural products was comparable to or even higher than that of the conventional fungicide. Specifically, in the case of *P. cinnamomi*, complete inhibition was achieved at concentrations of 187.5 μg·mL^−1^ and 78.1 μg·mL^−1^ for the non-conjugated extract and the conjugate complex with COS, respectively, whereas Fosetyl-Al exhibited only 12% inhibition at a concentration of 200 μg·mL^−1^. 

### 3.5. Comparison of Efficacy in Excised Stems

Concerning the activity of the COS—*G. lucidum* extract conjugate complex as a protective treatment against *P. cinnamomi*, a comparison with other treatments against *Phytophthora* spp. is presented in Table 7. Its efficacy was similar to that of non-conjugated *Q. ilex* aqueous ammonia extract [61], although it was tested on *Prunus amygdalus* × *P. persica* excised stems rather than on *Q. ilex* ones. The activity of the COS—*G. lucidum* extract conjugate complex was higher than those of non-conjugated *Sambucus nigra* L. flower ammonia extract [73] and the COS–*Quercus suber* L. aqueous ammonia bark extract conjugate complex [74], but these were tested against *Phytophthora cactorum* and *Phytophthora megasperma*, respectively, so the comparison should be made with caution.

## 4. Materials and Methods

### 4.1. Reagents and Fungal Isolates

Ammonium hydroxide (50% *v*/*v* aqueous solution) was purchased from Alfa Aesar (Ward Hill, MA, USA). Acetic acid (80% in H_2_O, purum grade) and potato dextrose agar (PDA) were supplied by Sigma Aldrich Química S.A. (Madrid, Spain). High molecular weight chitosan and Neutrase^TM^ 0.8 L enzymes were acquired from Hangzhou Simit Chem. and Tech. Co. (Hangzhou, China) and Novozymes A/S (Bagsværd, Denmark), respectively. Commercial *G. lucidum* used for vibrational spectra comparisons was purchased from MundoReishi Salud S.L. (Palencia, Spain). The commercial fungicide used as a positive control in the in vitro experiments, namely, Fosbel^®^ (fosetyl-Al 80%, reg. no. 25502; Probelte), was kindly provided by the Plant Health and Certification Service (CSCV) of the Gobierno de Aragón.

*Phytophthora cinnamomi* Nirenberg & O’Donnell was supplied by the Centro de Sanidad Forestal de Calabazanos (Villamuriel de Cerrato, Palencia, Spain); *Diplodia corticola* Phillips, Alves & Luque (CAA500 isolate) was kindly provided by the Biology Department of the Universidade do Minho (Braga, Portugal); while *Botryosphaeria dothidea* (Moug. ex Fr.) Ces. De Not. (ITACYL_F141) and *Dothiorella iberica* Phillips, Luque & Alves (ITACYL_F066) isolates were provided by the Instituto Tecnológico Agrario de Castilla y León (ITACYL, Valladolid, Spain). All isolates were supplied as subcultures on PDA and refreshed.

### 4.2. Collection of Samples

*Ganoderma lucidum* carpophores growing on *Q. ilex* trees were collected in October 2021 in *El Royal* farm, in El Tejado de Béjar, Salamanca, Spain (40°26′42.4″ N 5°33′09.4″ W). Specimens were identified and authenticated by Prof. Dr. B. Herrero-Villacorta (Departamento de Ciencias Agroforestales, ETSIIAA, Universidad de Valladolid) and voucher specimens are available at the herbarium of the ETSIIAA. Different specimens (*n* = 20) were thoroughly mixed to obtain composite samples, which were shade-dried, pulverized to a fine powder in a mill grinder, homogenized, and sieved (1 mm mesh).

### 4.3. Extraction Process, Preparation of Chitosan Oligomers, and Preparation of Conjugate Complexes

An aqueous ammonia extraction medium was chosen due to the woody texture of *G. lucidum* and to achieve the dissolution of polyphenols and other bioactive compounds of interest. Briefly, 67.3 g of *G. lucidum* carpophore powder was first digested in an aqueous ammonia solution (140 mL H_2_O + 20 mL NH_3_) for 2 h, then sonicated in pulsed mode (with a 2 min stop every 2.5 min) for 10 min using a probe-type ultrasonicator (model UIP1000hdT; 1000 W, 20 kHz; Hielscher Ultrasonics, Teltow, Germany), and then allowed to stand for 24 h. It was neutralized to pH 7 using acetic acid. Finally, the solution was centrifuged at 9000 rpm for 15 min, and the supernatant was filtered through Whatman No. 1 paper. The extraction yield was 4.2% (2.86 g). 

Aliquots of the extract were freeze-dried for attenuated total-reflectance Fourier-transform infrared (ATR-FTIR) spectroscopy and GC−MS characterization. For the latter, 25 mg of the lyophilized extract was resuspended in 5 mL of methanol (HPLC grade) to obtain a 5 mg·mL^−1^ solution, which was filtered before injection.

Chitosan oligomers were prepared using the method previously reported in [75], resulting in a solution with oligomers with a molecular weight of less than 2 kDa. 

The COS–*G. lucidum* carpophore extract conjugate complex was obtained by combining solutions (both at a concentration of 3000 μg·mL^−1^) in a 1:1 (*v*/*v*) ratio, followed by sonication for 15 min (five pulses lasting 3 min each to keep the temperature below 60 °C). The solution was freeze-dried for ATR-FTIR characterization to confirm the formation of the conjugate complex.

### 4.4. G. lucidum Characterization Procedures

The infrared vibrational spectra of the *G. lucidum* dried samples, as well as that of a commercial *G. lucidum* sample, were registered using a Thermo Scientific (Waltham, MA, USA) Nicolet iS50 FTIR spectrometer, equipped with an in-built diamond ATR system. The spectra were collected over the 400–4000 cm^−1^ range, with a 1 cm^−1^ spectral resolution, taking the interferograms resulting from co-adding 64 scans.

The aqueous ammonia extract of *G. lucidum* carpophores was analyzed by GC–MS at the Research Support Services (STI) at Universidad de Alicante (Alicante, Spain), using an Agilent Technologies gas chromatograph model 7890A coupled to a quadrupole mass spectrometer model 5975C. The chromatographic conditions were as follows: injection volume = 1 µL; injector temperature = 280 °C, in splitless mode; initial oven temperature = 60 °C, 2 min, followed by a ramp of 10 °C/min up to a final temperature of 300 °C, 15 min. The chromatographic column used for the separation of the compounds was an Agilent Technologies HP-5MS UI column with a length of 30 m, a diameter of 0.250 mm, and a film thickness of 0.25 µm. The mass spectrometer conditions were as follows: temperature of the electron impact source of the mass spectrometer = 230 °C; temperature of the quadrupole = 150 °C; ionization energy = 70 eV. The identification of components was based on a comparison of their mass spectra and retention time with those of the authentic compounds and by computer matching with the database of the National Institute of Standards and Technology (NIST11).

### 4.5. In Vitro Antifungal and Anti-Oomycete Activity

The antifungal and anti-oomycete activity of the *G. lucidum* carpophore extract and the conjugate complex with COS was examined using the poisoned food method. Aliquots of stock solutions were added to the PDA medium to produce final concentrations in the range of 15.62–1500 µg·mL^−1^. Mycelial plugs were transferred from the margin of one-week-old PDA cultures of *B. dothidea*, *D. corticola*, *D. iberica,* and *P. cinnamomi* to plates filled with the amended media. For each treatment and concentration combination, three plates were used, and each experiment was carried out twice. The untreated control consisted of replacing the extract with the solvent used for extraction in the PDA medium. Fosbel^®^ (fosetyl-Al 80%, reg. no. 25502; Probelte, Murcia, Spain) was used as a positive control. Additional controls, consisting of pure PDA medium and PDA with the lowest concentration of the treatment, were also included to confirm the absence of contamination. Radial mycelium growth was quantified by measuring the average of two perpendicular colony diameters for each replicate. Growth inhibition was estimated after incubation in the dark at 25 °C for one week, using the formula: ((*d_c_* − *d_t_*)/*d_c_*) × 100, where *d_c_* is the average colony diameter in the untreated control and *d_t_* is the average diameter of the treated colony. Effective concentrations (EC_50_ and EC_90_) were estimated using PROBIT analysis in IBM SPSS Statistics v.25 (IBM; Armonk, NY, USA). The degree of interaction was estimated using Wadley’s method [76].

### 4.6. Protection Tests on Artificially Inoculated Excised Stems

Given the restrictions that apply to in vivo assays involving *P. cinnamomi*, the efficacy of the most active treatment in the in vitro tests (i.e., COS−*G. lucidum* carpophore extract conjugate complex) was investigated by artificial inoculation of excised stems in controlled laboratory conditions. Inoculation was performed according to the procedure proposed by Matheron et al. [77], with modifications as described in [61,73,74]. Young stems (1.5 cm diameter) of healthy *Q. ilex* plants were cut into 10 cm-long sections using a sterilized grafting knife. The excised stem pieces were immediately wrapped in moistened sterile absorbent paper. In the laboratory, the freshly excised stem segments were first immersed in a 3% NaClO solution for 10 min, then in 70% ethanol for 10 min, and then thoroughly rinsed four times with distilled water, to avoid superficial contaminants in the tissue. Some of the stem segments (*n* = 15 for the positive control, and *n* = 15 for the negative control) were soaked for 1 h in distilled water to be used as controls, while the remaining stem segments were soaked for 1 h in aqueous solutions containing an appropriate amount of the conjugate complex to obtain MIC, MIC × 5, and MIC × 10 concentrations (*n* = 15 segments/concentration). A coadjuvant (Alkir^®^, 1% *v*/*v*) was added to all the solutions, including the control, to facilitate the moistening and penetration of the treatment into the bark. After soaking, the stem pieces were allowed to dry, and the bark was carefully removed with a scalpel to reveal the cambium. The bark was then placed on an agar Petri dish and, in the case of the positive control and treated samples, it was inoculated by placing a plug (diameter = 5 mm) from the margin of a one-week-old PDA culture of *P. cinnamomi* on the center of the inner surface of the bark. After inoculation, stem segments were incubated in a humid chamber for 4 days at 24 °C and 95–98% relative humidity. The efficacy of the treatments was evaluated by measuring the lengths of the cankers that developed at the inoculation sites. Finally, the oomycete was re-isolated from the lesions and morphologically identified to fulfill Koch’s postulates.

### 4.7. Statistical Analysis

The results from the in vitro mycelial growth inhibition and ex situ necrosis lengths were subjected to statistical analysis using one-way analysis of variance (ANOVA). Post hoc comparisons of means were conducted using Tukey’s test at a significance level of *p* < 0.05. Homogeneity and homoscedasticity requirements were checked using Shapiro–Wilk and Levene tests. The statistical analysis was performed using IBM SPSS Statistics v.25 software.

## 5. Conclusions

This study provides valuable insights into the composition and antimicrobial activity of an aqueous ammonia extract of *Ganoderma lucidum* carpophores. The GC-MS characterization revealed the presence of chemical constituents such as oleic acid and its methyl ester, 1,2,3,4-butanetetrol, monomethyl azelate, undecane, and palmitic acid and its methyl ester, which have demonstrated antimicrobial properties in previous studies. In vitro tests demonstrated significant anti-oomycete and antifungal activity of the *G. lucidum* extract, further enhanced upon combination with chitosan oligomers. In particular, conjugate complexes based on the extract exhibited notable efficacy against *Phytophthora cinnamomi*, a serious threat to *Quercus* spp., resulting in complete inhibition at 78.12 μg·mL^−1^, which was confirmed in ex situ bioassays on holm-oak-excised stems. These findings highlight the potential of *G. lucidum* as a natural alternative to synthetic fungicides for controlling plant diseases caused by oomycetes and fungi, and suggest its promise as a bioactive product for safeguarding *Quercus* spp. in the *dehesa* ecosystem.

## Figures and Tables

**Figure 1 plants-12-02271-f001:**
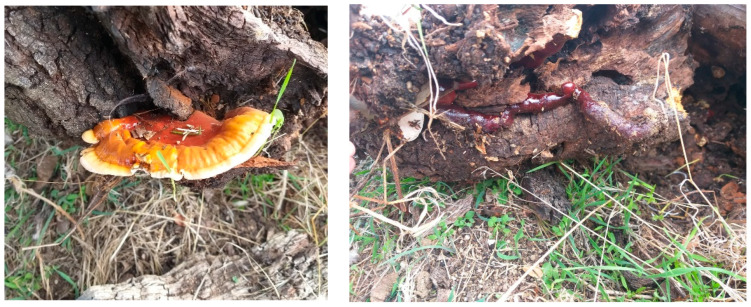
Detail view of a basidiocarp of *G. lucidum* (**left**) and its stipe (**right**).

**Figure 2 plants-12-02271-f002:**
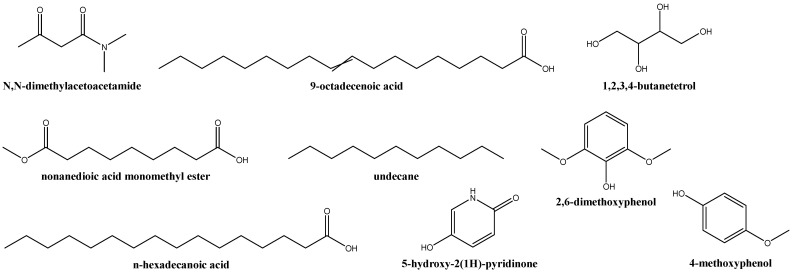
Chemical structures of the main chemical constituents identified in the aqueous ammonia extract of *G. lucidum* carpophores.

**Figure 3 plants-12-02271-f003:**
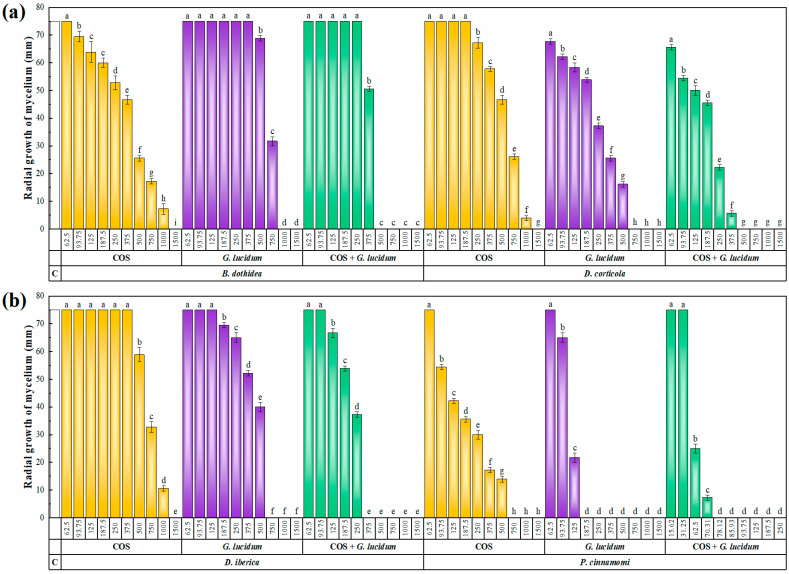
Mycelial growth inhibition achieved with chitosan oligomers (COS), aqueous ammonia extract of *G. lucidum* carpophores, and their conjugate complex (COS–*G. lucidum*) against (**a**) *B. dothidea* and *D. corticola*, and (**b**) *D. iberica* and *P. cinnamomi* at concentrations ranging from 62.5 to 1500 μg·mL^−1^ (or from 15.62 and 250 μg·mL^−1^ for COS–*G. lucidum* in the case of *P. cinnamomi*). The same letters above concentrations indicate that they are not significantly different at *p* < 0.05. Error bars represent standard deviations (*n* = 6). ‘C’ stands for the untreated control (i.e., PDA medium to which only the solvent used for extraction was added).

**Table 1 plants-12-02271-t001:** Main absorption bands (cm^−1^) in the infrared spectra of *G. lucidum* carpophore powder.

Wavenumber (cm^−1^)		Assignment
*G. lucidum* (This Study)	*G. lucidum* (Commercial)	
3290	3297	–OH and –NH stretch
2924	2922	–CH_2_ asymmetric stretching of alkyls (cutine, wax, pectin, amides)
2874		C–H stretching
(2183)		C–N bonding
(2148)		C=C stretching
(2047)		C–N bonds
(2018)		C–H stretching (polysaccharides)
1645	1634	C=O stretching (amides); C=C stretching; O–H deformation
1538		C–N bonds
1451		C–H bending
1374	1371	C–C asymmetrical stretching; phenolic OH groups; C–H (cellulose)
1203	1248	ketonic carbonyl group and C–N bonds
1153		C–C in plane (*β*-carotene); C–O–C asymmetric stretch (cellulose)
1036	1035	C–C stretching; C–N stretching; >C=O (ketonic) group
562		C–C out of plane bending; C–H rocking vibration
	526	C–C in-plane bending; COO^−^ rocking
452		C–C–C–C in-plane deformation

**Table 2 plants-12-02271-t002:** Main chemical constituents identified by GC-MS in *G. lucidum* carpophore aqueous ammonia extract.

Retention Time (min)	Peak Area (%)	Assignment	Qual
3.3124	28.279	Acetamide	90
5.1286	1.0155	5-(2-Chlorophenyl)-3-(1-piperidylmethyl)-1,3,4-oxadiazole-2(3H)-thione	59
6.1139	3.9257	Glycerin	78
6.1673	4.7722	1,2,3,4-Butanetetrol, [S-(R*,R*)]-	64
6.9863	2.1160	2-Propenethioamide, 3-(acetyloxy)-N,N-dimethyl-, (E)-	37
7.2123	1.29	2,5-Dimethyl-4-hydroxy-3(2H)-furanone	43
7.4078	0.9017	Fumaric acid, 3-methylbut-3-enyl tetradecyl ester	47
7.4790	0.7049	Tetrahydrofuran, 2-ethyl-5-methyl-	38
7.5442	1.70	2,5-Furandione, dihydro-3-methylene-	50
7.6511	2.1911	Mequinol	86
7.7995	4.4650	Undecane	42
8.5651	0.8559	4H-Pyran-4-one, 2,3-dihydro-3,5-dihydroxy-6-methyl-	62
9.4732	1.0981	Catechol	93
10.9867	0.8408	2-Methoxy-4-vinylphenol	64
11.0936	1.7093	N,N-Dimethylacetoacetamide	50
11.3312	1.86	N-Methoxy-2-carbaminoaziridine	49
11.4616	2.5489	Phenol, 2,6-dimethoxy-	97
11.8652	0.7987	DL-Proline, 5-oxo-, methyl ester	72
12.0491	2.5475	2(1H)-Pyridinone, 5-hydroxy-	64
13.2065	0.9025	Suberic acid monomethyl ester	64
13.2778	0.9487	Apocynin	81
13.5211	1.5205	Thiazole, 5-ethenyl-4-methyl-	35
13.6992	1.6385	Dodecanoic acid, methyl ester	98
14.1384	0.5720	Dodecanoic acid	96
14.1859	0.4162	Propenoic acid, 3-(1-ethyl-3-methyl-4-pyrazolyl)-	46
16.0317	1.2402	Methyl tetradecanoate	97
16.1208	0.8852	Ethanone, 1-(4-hydroxy-3,5-dimethoxyphenyl)-	96
16.7262	1.1287	Cyclohexanone, 5-methyl-2-(1-methylethyl)-, O-methyloxime, (2S-trans)-	38
18.1456	2.5903	n-Hexadecanoic acid ester	96
18.4889	2.0519	n-Hexadecanoic acid	95
19.8304	4.16	9-Octadecenoic acid, methyl ester	99
20.0678	0.9002	Methyl stearate	89
20.1865	3.8668	9-Octadecenoic acid, (E)-	96

Qual = Quality of resemblance.

**Table 3 plants-12-02271-t003:** Effective concentrations (EC, expressed in µg·mL^−1^) against *B. dothidea*, *D. corticola*, *D. iberica*, and *P. cinnamomi* of chitosan oligomers (COS), the aqueous ammonia extract of *G. lucidum* carpophores, and their conjugate complex (COS–*G. lucidum*). Synergy factors (SF) for the COS–*G. lucidum* extract conjugate complex were estimated according to Wadley’s method.

Pathogen	Treatment							
	COS		*G. lucidum*		COS–*G. lucidum*			
	EC_50_	EC_90_	EC_50_	EC_90_	EC_50_	SF	EC_90_	SF
*B. dothidea*	428.5	956.9	692.7	938.2	404.0	1.31	479.2	1.98
*D. corticola*	592.8	969.5	256.0	621.6	206.5	1.73	350.7	2.16
*D. iberica*	697.3	1201.7	476.4	703.8	249.6	2.27	345.6	2.57
*P. cinnamomi*	166.4	595.3	112.6	169.4	50.2	2.68	72.6	3.63

**Table 4 plants-12-02271-t004:** Mycelial growth inhibition achieved with Fosetyl-Al at the recommended dose (Rd = 2000 μg·mL^−1^) and at one tenth of the recommended dose (Rd/10 = 200 μg·mL^−1^) against the four phytopathogens under study.

Pathogen	Radial Growth of Mycelium (mm)		Inhibition (%)	
	Rd/10	Rd	Rd/10	Rd
*B. dothidea*	38.9	0	48.1	100
*D. corticola*	42.8	0	42.9	100
*D. iberica*	75.0	0	0	100
*P. cinnamomi*	65.5	0	12.7	100

The radial growth of the mycelium for the control (PDA only) was 75 mm. All mycelial growth values (in mm) are average values (*n* = 3).

**Table 5 plants-12-02271-t005:** Analysis of the differences in necrosis lengths between the treatments with a confidence interval of 95% (*p* < 0.0001).

Treatment	LS Means (Necrosis Length (mm))	Groups		
C+	40.467	A		
MIC	37.400	A		
MIC × 5	13.067		B	
MIC × 10	1.800			C
C−	0.000			C

C+: positive control (inoculated, no treatment); C−: negative control (not inoculated).

**Table 6 plants-12-02271-t006:** Activities reported in the literature for plant extracts against the four phytopathogens studied in this work.

Phytopathogen	Extraction Media	Plant	Efficacy	Ref.
*B. dothidea*	Ethanol 80%	Chinese herbal extract compound (*Scutellaria baicalensis*, *Syzygium aromaticum*, *Cinnamomum cassia*, *Gleditsia sinensis*, *Pogostemon cablin*, *Acorus calamus*, and *Camellia oleifera*, ratio 1.375:1.125:0.45:0.5:1.35:1.25:2.8)	IR = 85%, at 800 µg·mL^−1^	[60]
	Methanol 100%	*Hemizygia transvaalensis*	n.a.	[64]
		*Pearsonia aristata*	IR = n.a.–<40%, at 100,000 µg·mL^−1^	
		*Thesium burkei*	n.a.	
		*Alloteropsis semialata*	n.a.	
		*Smilax anceps*	n.a.	
		*Schrebera alata*	IR = n.a.–<40%, at 100,000 µg·mL^−1^	
		*Syncolostemon eriocephalus*	IR = <40%, at 100,000 µg·mL^−1^	
		*Eucomis autumnalis*	IR = 85%, at 100,000 µg·mL^−1^	
		*Mundulea sericea*	IR = <40%, at 100,000 µg·mL^−1^	
		*Brachylaena huillensis*	IR = <40%, at 100,000 µg·mL^−1^	
		*Lapholaena* sp.	IR = <40%, at 100,000 µg·mL^−1^	
	Methanol 95%	*Dolichos kilimandscharicus*	IR ≥ 60%, at 1000 µg·mL^−1^	[65]
		*Maerua subcordata*	IR < 50%, at 1000 µg·mL^−1^	
		*Phytolacca dodecandra*	IR < 50%, at 1000 µg·mL^−1^	
	Water	COS–*Equisetum arvense*	MIC = 750 µg·mL^−1^	[41]
		COS–*Urtica dioica*	MIC = 375 µg·mL^−1^	
*D. corticola*	Ethanol 50%	*Plantago major*	IR = 14.6%, at 2000 µg·mL^−1^	[66]
		*Medicago* sp.	IR = 60.9%, at 2000 µg·mL^−1^	[67]
		*Melilotus indicus*	IR = 16.7%, at 2000 µg·mL^−1^	
		*U. dioica*	IR = 34.1%, at 2000 µg·mL^−1^	
		*Medicago* sp., *M. indicus*, *P. major*, and *U. dioica*	IR = 15.8%, at 2000 µg·mL^−1^	
	Water or ethanol	*Rosmarinus officinalis*	IR = 52.2%, at 1500 µg·mL^−1^	[68]
	Ethanol 100%	*Cistus ladanifer*	IR = 38.75%, at 1000 µg·mL^−1^	[69]
	Ethanol 80%	*Musa* sp.	IR = 6–20%, at 750 µg·mL^−1^	[26]
		*Allium sativum*	IR = >50%, at 750 µg·mL^−1^	
		*Citrus lemon*	IR = 21–49%, at 750 µg·mL^−1^	
		*Citrus sinensis*	IR = 21–49%, at 750 µg·mL^−1^	
		*Allium cepa*	IR = 21–49%, at 750 µg·mL^−1^	
		*Punica granatum*	n.a.	
		*Solanum tuberosum*	IR = 21–49%, at 750 µg·mL^−1^	
		*Eucalyptus* sp.	IR = 6–20%, at 750 µg·mL^−1^	
		*Pinus* sp.	IR = 21–49%, at 750 µg·mL^−1^	
		*Olea europea*	IR = 6–20%, at 750 µg·mL^−1^	
*D. iberica*	Water	COS–*Equisetum arvense*	MIC = 750 µg·mL^−1^	[41]
		COS–*Urtica dioica*	MIC = 1000 µg·mL^−1^	
*P. cinnamomi*	Aqueous ammonia	*Quercus ilex* subsp*. ballota*	MIC = 78.12 µg·mL^−1^	[61]
	Ethanol 50%	*P. major*	IR = 32.2%, at 2000 µg·mL^−1^	[66]
	Ethanol 50%	*Medicago* sp.	IR = 21.5%, at 2000 µg·mL^−1^	[67]
		*M. indicus*	IR = 87.5%, at 2000 µg·mL^−1^	
		*U. dioica*	IR = 40%, at 2000 µg·mL^−1^	
		*Medicago* sp., *M. indicus*, *P. major*, and *U. dioica*	IR = 72.6%, at 2000 µg·mL^−1^	
	Water or ethanol	*R. officinalis*	IR = 33.9%, at 1500 µg·mL^−1^	[68]
	Ethanol 80%	*Musa* sp.	n.a.	[26]
		*A. sativum*	IR > 50%, at 750 µg·mL^−1^	
		*C. lemon*	IR = 21–49%, at 750 µg·mL^−1^	
		*C. sinensis*	IR = 21–49%, at 750 µg·mL^−1^	
		*A. cepa*	IR > 50%, at 750 µg·mL^−1^	
		*P. granatum*	n.a.	
		*S. tuberosum*	n.e.	
		*Eucalyptus* sp.	n.a.	
		*Pinus* sp.	IR = 21–49%, at 750 µg·mL^−1^	
		*O. europea*	n.a.	
	Water	*Larrea tridentata*	MIC_90_ = 1431 µg·mL^−1^	[62]
		*Flourensia cernua*	MIC_90_ = 193.4 µg·mL^−1^	
		*Agave lechuguilla*	MIC_90_ = 68,568 µg·mL^−1^	
		*Opuntia ficus-indica*	MIC_90_ = 121.7 µg·mL^−1^	
		*Lippia graveolens*	MIC_90_ = 4825 µg·mL^−1^	
		*Carya illinoensis*	n.a.	
		*Yucca filifera*	n.a.	
	Essential oil	*Salvia officinalis*	MIC > 1600 µg·mL^−1^	[63]
		*Salvia rosmarinus*	MIC > 1600 µg·mL^−1^	
		*Origanum vulgare*	MIC > 200 µg·mL^−1^	
		*Laurus nobilis*	MIC > 1600 µg·mL^−1^	
		*Coriandrum sativum*	MIC = 800 µg·mL^−1^	
		*Thymus vulgaris*	MIC = 200 µg·mL^−1^	
		*Mentha piperita*	MIC = 800 µg·mL^−1^	
		*Lavandula intermedia*	MIC = 1600 µg·mL^−1^	
		*Beilschmiedia miersii*	MIC = 300 µg·mL^−1^	[70]
	Methanol	*Arbutus unedo*	MIC = 5990 µg·mL^−1^	[71]
	Water	*P. granatum* cv. ‘Wonderful’	IR < 40%, at 10,000 µg·mL^−1^	[72]

COS: chitosan oligomers; IR: inhibition rate; IZ: inhibition zone; MIC: minimum inhibitory concentration; n.a.: no activity.

**Table 7 plants-12-02271-t007:** Protective treatments against *Phytophthora* spp. based on natural products.

Source of Excised Stems	Pathogen	Natural Product	Effectiveness	Ref.
*Quercus ilex*	*Phytophthora* *cinnamomi*	COS–*Ganoderma lucidum* ammonia carpophore extract conjugate complex	Full protection at 782 μg·mL^−1^	This work
*Prunus amygdalus* × *P. persica*	*P. cinnamomi*	*Q. ilex* subsp. *ballota* aqueous ammonia bark extract	Full protection at 782 μg·mL^−1^	[61]
	*Phytophthora* *cactorum*	COS–*Quercus suber* aqueous ammonia bark extract conjugate complex	Full protection at 3750 μg·mL^−1^	[74]
	*Phytophthora megasperma*	*Sambucus nigra* flower aqueous ammonia extract	Full protection at 1875 μg·mL^−1^	[73]

## Data Availability

The data presented in this study are available on request from the corresponding author.

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
