# Peer review of "Chemical Constituents and Antimicrobial Activity of a Ganoderma lucidum (Curtis.) P. Karst. Aqueous Ammonia Extract"

_plants, 2023, doi:10.3390/plants12122271_

Round 1
Reviewer 1 Report
The manuscript entitled “Chemical Constituents and Antimicrobial Activity of
Ganoderma lucidum (Curtis.) P. Karst. Aqueous Ammonia Extract” by Sánchez‐Hernández, et al gives an overview on the chemical investigation of Ganoderma lucidum by gas chromatography‐mass spectrometry. They further test the anti‐oomycete and antifungal activity against plant pathogens include Phytophthora cinnamomi, Botryosphaeria dothidea, Diplodia corticola, and Dothiorella iberica.
Key words: should be changed to be more representative to the work
Change “chemical species” to “chemical constituents” in the whole manuscript
What were the Positive controls used in the biological assays?
English should be checked by native English speaker
The authors did not isolate the compounds which could be responsible for activity, instead docking study should be done in order to predict the compound(s) that might be responsible for that.
The results section does not reach the current standards for publication.
References need update
The conclusions part is very poor, should be elaborated.
The manuscript entitled “Chemical Constituents and Antimicrobial Activity of
Ganoderma lucidum (Curtis.) P. Karst. Aqueous Ammonia Extract” by Sánchez‐Hernández, et al gives an overview on the chemical investigation of Ganoderma lucidum by gas chromatography‐mass spectrometry. They further test the anti‐oomycete and antifungal activity against plant pathogens include Phytophthora cinnamomi, Botryosphaeria dothidea, Diplodia corticola, and Dothiorella iberica.
Key words: should be changed to be more representative to the work
Change “chemical species” to “chemical constituents” in the whole manuscript
What were the Positive controls used in the biological assays?
English should be checked by native English speaker
The authors did not isolate the compounds which could be responsible for activity, instead docking study should be done in order to predict the compound(s) that might be responsible for that.
The results section does not reach the current standards for publication.
References need update
The conclusions part is very poor, should be elaborated.
Author Response
The manuscript entitled “Chemical Constituents and Antimicrobial Activity of Ganoderma lucidum (Curtis.) P. Karst. Aqueous Ammonia Extract” by Sánchez‐Hernández, et al gives an overview on the chemical investigation of Ganoderma lucidum by gas chromatography‐mass spectrometry. They further test the anti‐oomycete and antifungal activity against plant pathogens include Phytophthora cinnamomi, Botryosphaeria dothidea, Diplodia corticola, and Dothiorella iberica.
Q1. Keywords: should be changed to be more representative to the work
Response: In accordance with the journal guidelines, we have revised the keywords to comply with the requirement of including three to ten relevant keywords for indexing purposes. The selected keywords are specific to the article while being reasonably common within the subject discipline. We have referred to the keywords indicated in the special issue (available at https://www.mdpi.com/journal/plants/special_issues/Novel_Plant_Protect#keywords) as a reference.
The following changes have been made:
- 'Dehesa' has been replaced with 'dehesa ecosystem'.
- 'GC-MS' has been replaced with 'gas chromatography-mass spectrometry (GC-MS)'.
- 'Holm oak' has been replaced with 'Quercus ilex'.
- The scientific names of phytopathogens have been replaced with the more generic term 'phytopathogens'.
- We have added the following keywords: 'mushroom extracts', 'antifungal activity', 'anti-oomycete activity', 'natural products', and 'chitosan oligomers (COS)'.
- 'Reishi' has been retained, as it is the common name of Ganoderma lucidum, and using this term avoids repetition with the title.
These adjustments aim to enhance the article's discoverability and relevance within the journal's indexing system.
Q2. Change “chemical species” to “chemical constituents” in the whole manuscript
Response: ‘chemical species’ has been replaced with ‘chemical constituents’ throughout the manuscript, as suggested by the Reviewer.
Q3. What were the Positive controls used in the biological assays?
Response: We appreciate the reviewer's suggestion regarding the inclusion of a positive control in our experiments. We agree that having a positive control is important for evaluating the experimental technique. In our in vitro tests, we used Fosetyl-Al, a conventional synthetic fungicide widely used against Phytophthora spp. and fungi associated with grapevine trunk diseases (GTDs), as a comparison reference.
However, it is important to note that the primary purpose of our in vitro experiments was to measure the decrease in growth compared to the untreated control, representing the normal growth situation. The calculation of inhibition only requires the measurement of untreated control and treatment colonies. Typically, the positive control is not included in the reported data because comparing the extract with a conventional fungicide would not provide much informative value, as it would be a comparison between a complex mixture of chemicals and a pure chemical. Moreover, the reproducibility of our results with high precision and the concentration-dependent effects of the extract should suffice to validate our experimental technique.
Nevertheless, we acknowledge the reviewer's request, and in response, we have included data for Fosetyl-Al in the results section of the revised manuscript (Table 4). Additionally, a new subsection (3.4) on the comparison vs. the positive control has been included in the discussion section. We hope this will address the reviewer's concerns.
Q4. English should be checked by native English speaker
Response: We appreciate the Reviewer’s feedback and want to clarify that we have taken steps to ensure the quality of the language in the manuscript. In line with our usual practice, we engaged an educated native English speaker, Dr. Melissa Boyd, to proofread the manuscript. Although we do not always mention this in the acknowledgments, we consistently seek language assistance for our manuscripts to ensure their clarity and accuracy (as exemplified in Agronomy 2023, 13(2), 496; https://doi.org/10.3390/agronomy13020496).
Although the Reviewer did not specifically highlight any language-related issues, one of the co-authors, who possesses a C2-level certificate according to the CEFR, thoroughly reviewed the updated text. Additionally, we utilized Grammarly Pro and Writefull analysis tools to identify and address potential grammar mistakes. These measures were taken to ensure the manuscript's language quality and overall coherence.
Thank you for bringing this to our attention, and we are confident that these efforts have contributed to the manuscript's language proficiency.
Q5. The authors did not isolate the compounds which could be responsible for activity, instead docking study should be done in order to predict the compound(s) that might be responsible for that.
Response: Given the aim of our study, which is to evaluate the extract's efficacy as a biorational product rather than isolating specific phytochemicals, we believe that isolating compounds would have limited practical significance. As for the Reviewer's suggestion of conducting a docking study to identify the compound(s) responsible for the observed antimicrobial activity, we feel that utilizing computational methods goes beyond the scope of our study. We would like to maintain our focus on the overall efficacy of the extract. Furthermore, the antimicrobial activity of the major constituents has been well-documented in previous studies (as discussed in subsection 3.1). These studies have employed actual in vitro and in vivo tests, providing empirical evidence rather than relying on simulations. We thank the Reviewer for his/her understanding.
Q6. The results section does not reach the current standards for publication.
Response: Since the Reviewer did not provide any specific suggestions for improvement, we assume that the changes we made in response to the other Reviewers' comments have helped enhance the results section and the overall manuscript. We have carefully considered all the feedback received and made revisions accordingly to ensure the quality and clarity of the manuscript, paying particular attention to the results section.
Q7. References need update
Response: Taking into consideration that there are 27 references from the past 3 years (5 references to papers published in 2023, 8 references to articles published in 2022, and 14 references to articles published in 2021), we believe that the request to update the references is not necessary. These recent references provide valuable information for comparisons with the existing literature and ensure that the study incorporates up-to-date research findings. Concerning other older references, they are relevant for making comparisons with the existing literature. It is important to note that the publication year should not significantly affect the relevance or validity of the references. Restricting the bibliographical survey to a specific time frame, such as the past 5 years, could introduce bias and potentially overlook relevant studies published earlier. We have carefully considered the Reviewer's suggestion, but we believe that the current selection of references adequately supports the study's objectives and findings.
Q8. The conclusions part is very poor, should be elaborated.
Response: The conclusions section has been entirely rewritten: “This study provides valuable insights into the composition and antimicrobial activity of an aqueous ammonia extract of Ganoderma lucidum carpophores. The GC-MS characterization revealed the presence of chemical constituents such as oleic acid and its methyl ester, 1,2,3,4-butanetetrol, monomethyl azelate, undecane, and palmitic acid and its methyl ester, which have demonstrated antimicrobial properties in previous studies. In vitro tests demonstrated significant anti-oomycete and antifungal activity of the G. lucidum extract, further enhanced upon combination with chitosan oligomers. In particular, conjugate complexes based on the extract exhibited notable efficacy against Phytophthora cinnamomi, a serious threat to Quercus spp., resulting in complete inhibition at 78.12 μg·mL−1, which was confirmed in ex-situ bioassays on holm oak excised stems. These findings highlight the potential of G. lucidum as a natural alternative to synthetic fungicides for controlling plant diseases caused by oomycetes and fungi, and suggest its promise as a bioactive product for safeguarding Quercus spp. in the dehesa ecosystem.”
Reviewer 2 Report
Dear Authors,
The work presented in your manuscript can benefit from major revision. Main concern with the antifungal study, mostly due to the lack of positive control. Please find the detailed revision on my part attached.

The language must be improved, there is lack of organization and clarity in many sentences as well as some mistakes here and there. Main points highlighted in the attached PDF file.
Author Response
Dear Authors,
The work presented in your manuscript can benefit from major revision. Main concern with the antifungal study, mostly due to the lack of positive control. Please find the detailed revision on my part attached.
Q1. (Line 15) The abstract is lengthy and lacks clarity. It can benifit from some rephrasing. Especially emphasis on why the target fungus was investigated.
Response: The abstract has been rewritten. It now reads: “Mushroom extracts have shown potential as a source of new antimicrobial agents. This study investigates the chemical profile of an aqueous ammonia extract obtained from the carpophores of Ganoderma lucidum, which grows on Quercus ilex trees, and explores its valorization as a biorational. The major chemical constituents of the extract, identified through gas chromatography-mass spectrometry, include acetamide, oleic acid, 1,2,3,4-butanetetrol, monomethyl azelate, undecane, and palmitic acid. The anti-oomycete and antifungal activity of G. lucidum extract was evaluated against Phytophthora cinnamomi, the primary threat to Quercus spp. in the dehesa biome, as well as three Botryosphaeriaceae fungi. In vitro tests revealed minimum inhibitory concentration (MIC) values of 187.5 μg·mL−1 against P. cinnamomi and 187.5–1000 μg·mL−1 against the fungi. Furthermore, conjugation of the G. lucidum extract with chitosan oligomers (COS) synergistically enhanced its antimicrobial activity, resulting in MIC values of 78.12 and 375-500 μg·mL−1 against P. cinnamomi and the fungi, respectively. These MIC values are among the highest reported to date for natural products against these phytopathogens. Subsequent ex-situ testing of the COS-G. lucidum conjugate complex on artificially inoculated Q. ilex excised stems resulted in high protection against P. cinnamomi at a dose of 782 µg·mL−1. These findings support the potential utilization of this resource from the dehesa ecosystem to protect the holm oak, aligning with sustainable and circular economy approaches.”
Q2. (Line 18) Acetamide is a functional group, either mention the name of the acetamide identified or "acetamides" as in a group of acetamide containing molecules.
Response: We are referring to the chemical species with CAS No. 60-35-5, i.e., acetamide or ethanamide, not to the functional group.
Q3. (Lines 19-22) The sentence is very lengthy and lacks clarity, please break down into more digestible sentences.
Response: The sentence has been shortened and rephrased for clarity.
Q4.(Lines 19-22) Evaluate would be more fitting in an organic chemistry context.
Response: Ok.
Q5. (Line 34) The introduction is quite comprehensive and provides an excellent overview of the topic. However, it can benifit from some restructuring to be more concise with a clear flow of ideas.
For instance, the paragraph describing the extraction of metabolites should should precede the biological activity and major chemical constituents paragraph. Extraction, major chemical constituents, their biological activity.
Additionally, there is excessive details that are unrelated to the study in hand. Try to trim down irrelevant past research about the species.
Finally, I would advice against using a list format in the last paragraph for the aim of the study, as there are only two objectives.
Response: All the suggested changes have been attended to. The introduction has been restructured, unnecessary details about past research about the species have been deleted, and the list format in the last paragraph is no longer used.
Q6. (Lines 48-49) Not relevant
Response: Deleted.
Q7. (Lines 58-66) Natural products or metabolites.
Response: ‘Chemicals’ has been replaced with ‘metabolites’, as suggested by the Reviewer.
Q8. (Lines 58-66) Very lengthy and difficult to read sentence. Break down and trim the unnecessary information.
Response: All authorities for bacterial and fungal pathogens have been removed, keeping only the scientific names (binomial names), which should improve the readability. For consistency reasons, authorities have also been removed for all microbial scientific names throughout the manuscript.
Q9. (Line 67) Acetone doesn't produce the activity. Acetone extract does.
Response: No longer applicable. The sentence has been deleted as part of the changes made to the introduction section.
Q10. (Line 114) Aqueous ammonia extract must be mentioned here.
Response: Clarified. It now reads: “[…] G. lucidum carpophore aqueous ammonia extract […]”
Q11. (Line 115) Which one? since individual components are being discussed.
Response: NIST.11 database indicates ‘acetamide’ and CAS No. 000060-35-5, as shown below. It does not refer to the functional group but to the ethanamide chemical species. The text now reads: “[…] acetamide or ethanamide (28.3%), […]”
Pk# RT Area% Library/ID Ref# CAS# Qual
_____________________________________________________________________________
1 3.312 28.28 C:\Database\NIST11.L
Acetamide 241 000060-35-5 90
Q12. (Line 129) The results are quite well presented here. However, I find the lack of positive or negative controls rather concerning. Additionally, it is not obvious if these experiments were run in triplicates?
It is of utmost importance to run these types of experiments in the presence of a positive control (a highly active antimicrobial agent). Additionally, the triplicate run increases the reliability of the data tremendously. Unless there is a compelling reason for why this is better done without control, I believe it must be repeated.
Response: As noted in subsection 4.5, in the in vitro tests, “the control consisted of replacing the extract with the solvent used for extraction in the PDA medium”. We had used the usual wording for the food poisoned method (see, for instance, Methods for in vitro evaluating antimicrobial activity: A review - PMC (nih.gov)), in which ‘control’ refers to the ‘untreated’ medium, amended only with the solvent used for extraction, as in [Antifungal activity of plant extracts against dermatophytes - PubMed (nih.gov)]). To avoid confusion to readers who are not familiar with this testing procedure, we have now clarified that we were referring to the ‘untreated control’. In section 2.3, in the Figure 3 caption, we have now specified that: “[…] ‘C’ stands for the untreated control (i.e., PDA medium to which only the solvent used for extraction was added).”. With regard to negative controls, they were indeed included (we tested pure PDA and PDA with the lowest concentration of the extract, 3 plates each, with 2 repeats), to confirm the absence of contamination. We have added a line to the materials and methods section to explain that we also had those negative controls.
As for the comment on the replicates, as explained in subsection 4.5, “[…] For each treatment and concentration combination, three plates were used, and each experiment was carried out twice. […]”. Consequently, the Figure 3 caption has also been updated to indicate that n = 6: “[…] ·Error bars represent standard deviations (n = 6). […]”.
Concerning the inclusion of a positive control in our experiments, we agree that having a positive control is important for evaluating the experimental technique. In our in vitro tests, we used Fosetyl-Al, a conventional synthetic fungicide widely used against Phytophthora spp. and fungi associated with grapevine trunk diseases (GTDs), as a comparison reference.
However, it is worth noting that the primary purpose of our in vitro experiments was to measure the decrease in growth compared to the untreated control, representing the normal growth situation. The calculation of inhibition only requires the measurement of untreated control and treatment colonies. Typically, the positive control is not included in the reported data because comparing the extract with a conventional fungicide would not provide much informative value, as it would be a comparison between a complex mixture of chemicals and a pure chemical. Moreover, the reproducibility of our results with high precision and the concentration-dependent effects of the extract should suffice to validate our experimental technique.
Nevertheless, we acknowledge the reviewer's request, and in response, we have included data for Fosetyl-Al in the results section of the revised manuscript (Table 4). Additionally, a new subsection (3.4) on the comparison vs. the positive control has been included in the discussion section. We hope this will address the reviewer's concerns.
Q13. (Line 136) Ranging from XX to XX.
Response: Corrected. The sentence now reads: “[…] with minimum inhibitory concentrations (MICs) ranging from 187.5 to 1000 μg·mL−1 and from 750 to 1500 μg·mL−1, respectively […]”.
Q14. (Lines 178-185) Rather confusing sentence. Rewrite for clarity to highlight the relationship between the acetamides found in table 2 and carboximidic acids. Also which carboximidic acid was identified before? or just carboximidic acids as a group of compounds?
Response: The comment on carboximidic acids has been removed to avoid potential confusion. The indicated sentence has been rewritten in a clearer manner. It now reads: “Among the list of compounds presented in Table 2, acetamide or ethanamide has been previously identified in red beetroots (Beta vulgaris var. rubra) and Clerodendrum infortunatum L. leaves [21]. It has also been found in extracts from Larrea divaricata Cav., Picea pungens Engelm., and Sequoiadendron giganteum (Lindl.) Buchholz. The presence of acetamide in the extract may be attributed to the partial hydrolysis of N,N-dimethylacetoacetamide, which was also identified in the extract. Alternatively, it could originate from N-(3-methylbutyl)acetamide or N(2-phenylethyl)acetamide, which are common components of fresh wild mushrooms [22]. However, it is worth noting that the presence of acetamide in the extract may be an artifact resulting from the extraction procedure, as it can also be formed through the de-composition of ammonium acetate. Ammonium acetate is generated by neutralizing excess ammonia in the extract with acetic acid. It is important to mention that acetamide-containing compounds are widely used as herbicides in agriculture [23], and several acetamide derivatives containing compounds have been reported to act as an-timicrobial agents [24].”
Q15. (Line 186) It is more accurate to write "acetamide containing compounds" instead of acetamides here. Because some compounds having this functional group are herbicides and not all acetamides are.
Response: ‘acetamide derivatives’ has been replaced with ‘acetamide containing compounds’, following the Reviewer’s suggestion.
Q16. (Lines 215-216) It is irrelevant to mention undecan-3-one, when talking about an undecane. It falls under ketones and not alkanes like undecane.
Response: The indicated sentence has been deleted. We now simply indicate that “There is no clear information available on the mechanism of action of undecane as an antimicrobial agent.”
Q17. (Line 233) What does metabolite mean here? as in pyriproxyfen is metabolized in the system to 5-hydroxy-2(1H)-pyridinone?
Response: The original statement came from a work by Yoshino et al. (https://pubs.acs.org/doi/abs/10.1021/jf00058a024) on the metabolism of Pyriproxyfen in rats and mice. It has been removed from the revised version of the manuscript, given that the information was not relevant.
Q18. (Lines 253-255) This is why a positive control is important drawing any conclusions or comparisons is entirely dependant on weather your experiment is valid.
Response: See response to Q12.
Q19. (Lines 311-316) May not be appropriate in the methods section. Delete or mention in the discussion instead.
Response: The indicated lines have been deleted.
Q20. (Lines 364-366) The negative control data should be reported in the text?
Response: As mentioned in the previous response to Q12, the data for the ‘untreated control' can be found in Figure 3, specifically in the leftmost column. In these experiments, the fungal taxa were grown on Petri dishes containing PDA medium amended only with the solvent, and the growth of the mycelium was monitored until it reached the border of the dish, which corresponds to a radial growth dc = 75 mm.
Regarding the other controls, it is worth noting that we did not observe any contamination in the PDA-only medium or in the PDA medium with the lowest concentration of the treatments (15.62 μg·mL−1). This observation is commonly expected and is generally taken for granted in such experiments. Therefore, we did not explicitly mention it in the results section as it is considered a standard control and did not require a separate repetition of the experiment.
Q21. (Line 407) Must be rephrased for clarity and to properly highlight the importance of the work carried out. There is also lack of organization, begin with a clear statement of research problem, identify the issue researched and how was it tackled. Be precise in doing so, avoid redundancy.
Response: The conclusions section has been entirely rewritten according to the Reviewer’s instructions: “This study provides valuable insights into the composition and antimicrobial activity of an aqueous ammonia extract of Ganoderma lucidum carpophores. The GC-MS characterization revealed the presence of chemical constituents such as oleic acid and its methyl ester, 1,2,3,4-butanetetrol, monomethyl azelate, undecane, and palmitic acid and its methyl ester, which have demonstrated antimicrobial properties in previous studies. In vitro tests demonstrated significant anti-oomycete and antifungal activity of the G. lucidum extract, further enhanced upon combination with chitosan oligomers. In particular, conjugate complexes based on the extract exhibited notable efficacy against Phytophthora cinnamomi, a serious threat to Quercus spp., resulting in complete inhibition at 78.12 μg·mL−1, which was confirmed in ex-situ bioassays on holm oak excised stems. These findings highlight the potential of G. lucidum as a natural alternative to synthetic fungicides for controlling plant diseases caused by oomycetes and fungi, and suggest its promise as a bioactive product for safeguarding Quercus spp. in the dehesa ecosystem.”
Round 2
Reviewer 1 Report
All comments have been addressed
Reviewer 2 Report
I would like to congratulate the authors on the work done. And The manuscript improvement in such short notice.
I have no further comments.